# Comparative Evaluation of Breast Ductal Carcinoma Grading: A Deep-Learning Model and General Pathologists’ Assessment Approach

**DOI:** 10.3390/diagnostics13142326

**Published:** 2023-07-10

**Authors:** Maria Magdalena Köteles, Alon Vigdorovits, Darshan Kumar, Ioana-Maria Mihai, Aura Jurescu, Adelina Gheju, Adeline Bucur, Octavia Oana Harich, Gheorghe-Emilian Olteanu

**Affiliations:** 1Bihor County Clinical Emergency Hospital, Gh. Doja Street No. 65, 410169 Oradea, Romania; maria.koteles@yahoo.com; 2Center for Research and Innovation in Personalized Medicine of Respiratory Diseases, “Victor Babes” University of Medicine and Pharmacy, Timisoara Eftimie Murgu Sq. No. 2, 300041 Timisoara, Romania; olteanu.gheorghe@umft.ro; 3Victor Babes Institute of Pathology—Next Generation Pathology Research Group, Splaiul Independenţei 99-101, 050096 Bucharest, Romania; 4Aiforia Technologies PLC, 00150 Helsinki, Finland; darshan.kumar@aiforia.com; 5Department of Microscopic Morphology-Morphopatology, ANAPATMOL Research Center, “Victor Babes” University of Medicine and Pharmacy, 300041 Timisoara, Romania; ioana.mihai@umft.ro (I.-M.M.); jurescu.aura@umft.ro (A.J.); 6Emergency County Hospital Deva, Bulevardul 22 Decembrie 58, 330032 Deva, Romania; gheju.adelina@umft.ro; 7Department of Microscopic Morphology, Discipline of Histology, “Victor Babes” University of Medicine and Pharmacy, Timisoara Eftimie Murgu Sq. No. 2, 300041 Timisoara, Romania; adebucur@yahoo.com; 8Department of Functional Sciences, “Victor Babes” University of Medicine and Pharmacy, Timisoara Eftimie Murgu Sq. No. 2, 300041 Timisoara, Romania; harich.octavia@umft.ro; 9Faculty of Pharmacy, “Victor Babes” University of Medicine and Pharmacy, Eftimie Murgu Square No. 2, 300041 Timisoara, Romania; 10Research Center for Pharmaco-Toxicological Evaluations, Faculty of Pharmacy, “Victor Babes” University of Medicine and Pharmacy, Eftimie Murgu Square No. 2, 300041 Timisoara, Romania; 11Center of Expertise for Rare Lung Diseases, Clinical Hospital of Infectious Diseases and Pneumophthisiology “Dr. Victor Babes” Timisoara, Gh. Adam Street No. 13, 300310 Timisoara, Romania

**Keywords:** breast cancer, infiltrating ductal adenocarcinoma, deep-learning model, convolutional neural network-based algorithm, artificial intelligence in cancer grading, observer agreement (AI and pathologists)

## Abstract

Breast cancer is the most prevalent neoplasia among women, with early and accurate diagnosis critical for effective treatment. In clinical practice, however, the subjective nature of histological grading of infiltrating ductal adenocarcinoma of the breast (DAC-NOS) often leads to inconsistencies among pathologists, posing a significant challenge to achieving optimal patient outcomes. Our study aimed to address this reproducibility problem by leveraging artificial intelligence (AI). We trained a deep-learning model using a convolutional neural network-based algorithm (CNN-bA) on 100 whole slide images (WSIs) of DAC-NOS from the Cancer Genome Atlas Breast Invasive Carcinoma (TCGA-BRCA) dataset. Our model demonstrated high precision, sensitivity, and F1 score across different grading components in about 17.5 h with 19,000 iterations. However, the agreement between the model’s grading and that of general pathologists varied, showing the highest agreement for the mitotic count score. These findings suggest that AI has the potential to enhance the accuracy and reproducibility of breast cancer grading, warranting further refinement and validation of this approach.

## 1. Introduction

Breast cancer is the most frequently diagnosed neoplasia among women, with a woman receiving this diagnosis every 18 s [1]. Early and accurate diagnosis is paramount for the effective treatment of this disease. However, a significant challenge that affects patient outcomes is the high degree of subjectivity involved in the histological grading of infiltrating ductal adenocarcinoma of the breast (DAC-NOS), which results in low reproducibility among pathologists. The traditional Nottingham grading system has been in use for over 50 years, translating tubule formation, nuclear variation, and mitotic activity into a numerical representation [2,3,4,5,6,7]. Despite its longevity, this system suffers from a degree of interobserver variability due to its reliance on the subjective interpretation of observers [8,9]. With the advent of digital pathology and whole slide imaging (WSI), the potential for improved image analysis has been recognized [10,11,12]. Additionally, the rise of artificial intelligence (AI) and machine learning (ML) techniques offers the potential to increase the accuracy of tumor grading, thereby reducing the subjectivity and variability inherent in the current system [13,14,15,16,17,18,19,20]. Our study aims to explore this potential further by developing and training a convolutional neural network-based algorithm (CNN-bA) to detect and quantify all three aspects of the Nottingham grading scheme. We then compare the results of our AI model with those produced by general pathologists (not specialized breast pathologists/thoracic pathologists) grading DAC-NOS, in an effort to evaluate the feasibility and efficiency of AI in enhancing the accuracy and reproducibility of breast cancer grading.

## 2. Materials and Methods

### 2.1. Dataset

The dataset comprised 100 cases of ductal adenocarcinoma of the breast (DAC-NOS) that were chosen randomly from the TCGA-BRCA WSI dataset. All of the selected WSI belonged to patients that were female, aged between 27 and 90, and all tumor stages were included. Other breast cancer types were excluded from the study. 10 training slides were selected (Appendix A) and used to train a CNN-bA to detect and quantify the components for DAC-NOS grading, these training slides were chosen from the TCGA-BRCA WSI dataset and were not included in the 100 test WSI. The WSIs were selected by an experienced thoracic pathologist (G.E.O).

### 2.2. Nottingham Grading by Pathologists

For an unequivocal assessment of the elements of NGS: tubule formation, nuclear pleomorphism, and mitotic count as well as final grading, the four pathologists that represented the human arm (HA) part of the study were trained in an in-person (independently of each other) instruction session to grade using the NGS all the WSI from our dataset using QuPath version 0.2.3 following a standardized methodology (Figure 1) [21]. No information was shared between the pathologists and there was no access to other clinical or pathological data. We defined a square-shaped region of interest (ROI) with a total surface area of 0.374 mm^2^, which is equal to the surface area of 10 high-power fields on a microscope with a field diameter of 0.66 mm. The ROIs were manually and independently chosen by each pathologist for each WSI from the dataset.

### 2.3. Machine Learning Approach

The model utilized in this study was developed using Aiforia Create (Version 5.5, Aiforia Technologies Plc). It comprised multiple independent yet nested convolutional neural networks (CNNs) that operated sequentially. Each CNN focused exclusively on the pixels passed to it by the preceding CNN, enabling the detection of specific objects or areas of interest, much like how pathologists analyze images (Figure 2). For the training of the CNN-bA, manual segmentation of the 10 training whole-slide images (WSI) was conducted by a thoracic pathologist (G.E.O). It’s important to note that these 10 training WSIs were distinct from the 100 test WSIs and were solely used for AI training purposes. The manual segmentation facilitated the extraction of all tissue and cell features necessary for the subsequent analysis using next-generation sequencing (NGS) (Figure 3). To address the sparsity of our training dataset, we incorporated Aiforia’s image augmentation features. This technique has demonstrated effectiveness in enhancing the model’s capacity to generalize to novel, unseen data while reducing input variables. These features included regions of normal tissue, regions of tumor tissue, tubular tumor component, and solid tumor component, object detection was used for pleomorphism and mitosis. The following parameters i.e., training procedure (TP) and image augmentations (IA), (Appendix A), were used for the layer of tissue, normal tissues vs. tumor tissue, tumor components (solid components and tubule formation), and tumor cells (pleomorphism and mitosis) as follows: 

**For the tissue layer, TP**-weight decay: 0.0001, mini-batch size: 80, mini-batches per iteration: 20, iterations without progress: 750, initial learning rate: 0.1. **IA**-scale (min/max): −10/10, aspect ratio: 10, maximum shear: 10, luminance (min/max): −10/10, contrast (min/max): −10/10, maximum white balance change: 5, noise in levels: 5, JPG compression quality (min/max): 40/60, blur maximum pixels: 1 px, JPG compression percentage: 0.5%, blur percentage 0.5%, rotational angle (min/max) −180/180 by flipping.

**For the normal tissues vs. tumor tissue, TP**-weight decay: 0.0001, mini-batch size: 40, mini-batches per iteration: 20, iterations without progress: 750, initial learning rate: 0.1. **IA**-scale (min/max): −10/10, aspect ratio: 10, maximum shear: 10, luminance (min/max): −10/10, contrast (min/max): −10/10, maxi-mum white balance change: 5, noise in levels: 5, JPG compression quality (min/max): 40/60, blur maximum pixels: 1 px, JPG compression percentage: 0.5%, blur percentage 0.5%, rotational angle (min/max) −180/180 by flipping.

**For the tumor components (solid components and tubule formation), TP**-weight decay: 0.0001, mini-batch size: 20, mini-batches per iteration: 20, iterations without progress: 750, initial learning rate: 0.1. **IA**-scale (min/max): −20/20, aspect ratio: 20, maximum shear: 20, luminance (min/max): −20/20, contrast (min/max): −20/20, maximum white balance change: 5, noise in levels: 5, JPG compression quality (min/max): 40/60, blur maximum pixels: 1 px, JPG compression percentage: 0.5%, blur percentage 0.5%, rotational angle (min/max) −180/180 by flipping.

**For the tumor cells (pleomorphism and mitosis), TP**-weight decay: 0.0001, mini-batch size: 100, mini-batches per iteration: 20, iterations without progress: 750, initial learning rate: 0.1. **IA**-scale (min/max): −1/1, aspect ratio: 1, maximum shear: 1, luminance (min/max): −1/1.01, contrast (min/max): −1/1.01, maximum white balance change: 1, noise in levels: 0, JPG compression quality (min/max): 40/60, blur maximum pixels: 1 px, JPG compression percentage: 0.5%, blur percentage 0.5%, rotational angle (min/max) −180/180 by flipping.

The CNN-bA was then trained using these features. For model evaluation, the breast pathologist selected a 0.374 mm^2^ tumor tissue ROI on each WSI. The model then performed the segmentation of the ROI. The percentage of tubule formation was calculated by dividing the total surface area of tubular structures by the total surface area of the tumor. The model detected the mitotic count in the ROI (equal in surface to 10 HPF). The pleomorphic cell counts for each WSI were gathered and their distribution was divided into thirds. The bottom third was defined as minimal pleomorphism, the middle third as moderate pleomorphism, and the upper third as marked pleomorphism. The grading of each WSI on the ROI was then performed. Interobserver variability was assessed between the pathologists and a comparison was made between the CNN-bA grading and the HA using the ROI analyzed as the data evaluated (comparator) between the CNN-bA and the HA.

### 2.4. Statistical Analysis

The pair-wise interobserver agreement was measured by calculating Cohen’s κ coefficient for the pathologists and the machine learning model. The levels of agreement were defined as follows: slight (<0.2), fair (0.21–0.40), moderate (0.41–0.60), good (0.61–0.8), and very good (0.81 to 1.00). Data processing and statistical analysis were performed using Microsoft Excel (2021, Microsoft Corporation) and Python (version 3.10, Python Software Foundation).

## 3. Results

### 3.1. Pathological Description of the Data Set Used for HA and AI Evaluation

Of the 100 WSI used for evaluation, 18 had no NGS assigned. All the lesions were diagnosed as infiltrating ductal carcinoma, NOS. Out of the 82 graded tumors, 47 (57.3%) were grade 3, 25 (30.4%) were grade 2, and 10 (12.2%) were grade 1. Most tumors were stage II, regardless of NGS. All stage III tumors had an NGS of 3, all data was extracted from the pathology reports from the TCGA-BRCA data (Appendix A). The association between staging and NGS in the evaluation dataset can be seen in Table 1.

For grade 1 and grade 2 tumors, a tubule formation score of 2, a nuclear pleomorphism score of 2, and a mitotic activity score of 1 were the most common. For grade 3 tumors, a score of 3 was the most common for each component. The association of the grading component scores and overall NGS is illustrated in Table 2.

### 3.2. Pathological Description of the Data Set Used for AI Model/Deep Learning-Based Model Training

Out of the 10 WSI used for training the deep learning (DL) model, 7 had NGS grading. All the tumors were diagnosed as infiltrating ductal carcinoma, NOS. Out of these 7 tumors, 3 were grade 1, 1 was grade 2, and 3 were grade 3. One of the tumors was stage I, 1 was stage IA, 1 tumor was IIA, 3 tumors were stage IIB, and 1 tumor was stage IIIA all data was extracted from the pathology reports from the TCGA-BRCA data, Table 3 and Appendix A.

### 3.3. Performance and Comparison of the HA and DL

Interobserver agreement for tubule formation was moderate (Cohen’s κ = 0.41–0.49) with two exceptions that presented only fair agreement, κ = 0.36 and k = 0.37 respectively (Figure 4). Regarding nuclear pleomorphism, overall interobserver agreement ranged from slight to fair corresponding to Cohen’s κ = 0.096–0.24 (Figure 4). The lowest agreement between pathologists was achieved for the mitotic count score, with values of Cohen’s κ ranging between 0.0054–0.29 (Figure 4).

For overall NGS grading, results showed multiple levels of agreement, from slight to moderate, with the highest agreement present between pathologists P2 and P4 (κ = 0.5) and the lowest agreement between P1 and P2, with a Cohen’s κ of 0.075 (Figure 4).

After calculating the interobserver agreement between the pathologists and the AI model, the highest agreement was achieved for the mitotic count score, with levels of agreement varying from slight to fair (Cohen’s κ = 0.016–0.24) (Figure 4). For tubule grading, the maximum value of Cohen’s κ was 0.19. Agreement concerning nuclear pleomorphism was the lowest, with a maximum Cohen’s κ of 0.036. Regarding the NGS agreement, P1 achieved the largest value of the interobserver agreement (Cohen’s κ of 0.21), while only slight levels of agreement were seen between DL and P2, P3 and P4 (κ = 0.11–0.16).

A comparative view of the NGS component scores and overall grading between the AI model and human pathologists (Figure 5). It is evident from the graph that the AI model tends to assign higher grades more frequently, particularly in severe cases. However, in instances of lower grade detections, the AI model’s grading aligns with the average human assessment, indicating a balanced approach. The figure underlines the AI model’s potential in identifying high-risk cases, alongside its ability to approximate human expertise in less severe cases.

HA grading from all 4 pathologists of all 100 WSI per histological component can be found in (Appendix A).

### 3.4. Performance of the DL Model

The total training time needed for the AI model used for the grading was 17 h, 34 min, 44 s, and 19,000 iterations. The results from the verification of the AI model used can be found in Table 4, and the data extraction example and visualization from the AI analysis are shown in Figure 6.

## 4. Discussion

Being one of the most frequent cancer types, ductal carcinoma of the breast has been studied over decades alongside the Nottingham grading system, the means to a worldwide consensus regarding classification [23,24]. While NGS relies mostly on translating recognized features into an ordinal scale, the interobserver disagreement between pathologists seems inescapable due to subjectivity. The purpose of this study was to address this issue and try to get a better understanding of how artificial intelligence may be used in the future to decrease or even eliminate subjectiveness when it comes to NGS. [25]. Previous research papers that tended to focus on interobserver variability had the disadvantage of using glass slides and different types of microscopes [26]. Longacre et al. [27] suggested that the evaluation of one of the key elements of NGS may have been imprecise due to slide processing and ocular lens. We were able to eliminate this kind of shortcoming by working with digital slides. The DL model was prepared by training 10 WSI, from which 3 had no grading, 3 were grade 1, one of them was grade 2 and the other 3 were grade 3. To mimic how a human observer would grade the images, a thoracic pathology expert performed the segmentation of the 10 WSI. Regions of normal tissue, tumoral tissue, solid tumor component, and the three features of NGS, tubule formation, nuclear pleomorphism, and mitotic count were all separately trained and analyzed, for final grading at the end. Whereas past researchers have only used AI to study individual components of the grading system, we were able to assess all three, to obtain a final grading, which was then compared with the HA [28,29]. One of the limitations concerning this study is that none of the four pathologists partaking in the final grading are breast pathology experts. Each of them has been practicing general pathology for several years, but since it has been suggested that pathologists not specialized in diseases of the breast tend to underscore, there is a possibility that this could have influenced the results we obtained [30]. Another limitation of our study is that the ROIs evaluated by the AI model were not identical to the ROIs evaluated by the pathologists, this discrepancy in the ROIs introduces a potential limitation, as the AI model was focused on the ROI chosen by the thoracic pathologist. The variations in ROI selection could have influenced the results and contributed to the observed differences in agreement levels between the AI model and the pathologists.

Our research is consistent with previous results found in the literature, with the lowest agreement between pathologists being about the mitotic count score. This can be explained by the fact that there was no guidance offered regarding which area of the tumor to be examined, therefore each pathologist was instructed to choose on their consideration. The present results regarding the overall NGS score are consistent with Delides et al.’s work [31], which showed a low interobserver agreement. One of the most recent studies on this subject, conducted by Mantrala et al. [9], offered slightly different results with a better level of agreement between AI and pathologists. The reason for that is probably the experience of the six observers in the field of breast pathology. Contrary to our expectations, there were only slight levels of agreement between our DL model and three of the pathologists, and only one showed a fair level of agreement. Despite these unanticipated results, our study is one of the very few ones that analyses the comparison between the human grading of breast ductal carcinoma and the deep-learning model on WSI. In our study, the agreement levels ranged from slight to fair, which suggests that the AI model is somewhat able to replicate the decisions made by human pathologists, but there is still a significant amount of variability. This could be due to several factors, including the complexity of the grading task, the variability in human grading, and the limitations of the AI model itself. The fact that there is some level of agreement is promising, as it suggests that AI has the potential to assist in this task. However, the relatively low levels of agreement also indicate that there is still a lot of room for improvement. Moreover, this research has offered a starting point for further research regarding the possibility of using DL and CNNs as an aid for pathologists, when it comes to NGS. The AI model may need to be trained on a larger or more diverse set of slides, or it may need to incorporate more complex features or algorithms. Alternatively, it might be beneficial to focus on improving the consistency of human grading, for example through more standardized training or guidelines. In any case, these results suggest that AI has the potential to play a role in the grading of breast cancer, but more research and development are needed to fully realize this potential. The most important contribution of this study may be that it raises a variety of questions regarding the utility of AI mechanisms and the importance of training AI on a larger number of slides. Furthermore, this research can be seen as the first step towards eliminating bias and interobserver variability, offering a direction headed for improving efficiency and unanimity in grading systems.

## 5. Conclusions

Our study illustrates the potential of AI in addressing the significant challenge of subjectivity and interobserver variability within the Nottingham grading system for breast ductal carcinoma. We trained a DL model on WSIs to evaluate all three components of the grading system, marking a departure from previous studies that examined these components individually. Although our study had limitations, such as the lack of specialized breast pathology expertise among the participating pathologists, our findings generally corroborate existing literature. Of note, the lowest agreement was found on the mitotic count score. Despite expectations, the agreement between our DL model and the pathologists’ assessments ranged from slight to fair, indicating the need for further refinement of the model and potentially the need for more experienced observers in breast pathology. Nonetheless, our study contributes meaningfully to the growing body of research examining the application of AI in medical grading systems. It suggests that training AI on a larger number of slides could potentially eliminate bias and interobserver variability, thereby improving efficiency and consensus in grading systems. Future research should consider involving breast pathology specialists and employing a larger, more diverse set of slides for AI training to further refine and validate our findings.

## Figures and Tables

**Figure 1 diagnostics-13-02326-f001:**
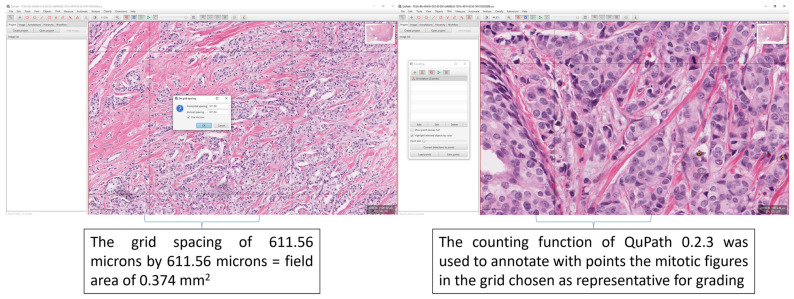
Extracts from the hands-out given to the grading pathologists after the training. The examples highlight how to perform the grid spacing in QuPath 0.2.3 to allow a grid of 611.56 microns by 611.56 microns, which equals a field area of 0.375 mm^2^ [22] for grading. Also highlighted is the counting function of QuPath 0.2.3 which was used to train the pathologists.

**Figure 2 diagnostics-13-02326-f002:**
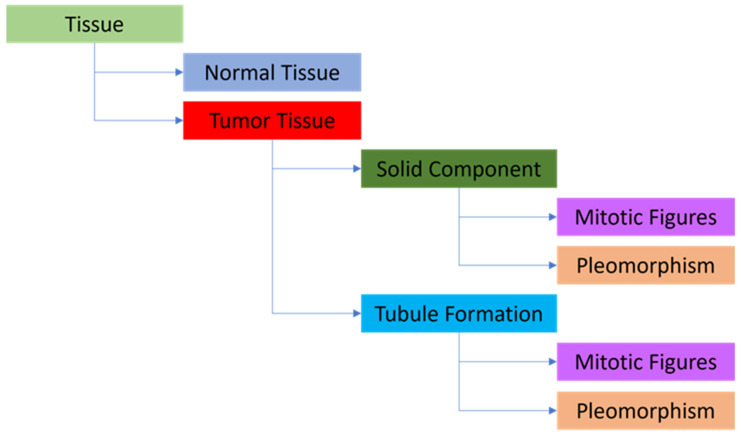
The layered structure is used for the AI model. Tissue detection—parent layer, Normal Tissue and Tumor Tissue—child layers, Solid Component and Tubule Formation (i.e., gland formation) —child layers, Mitotic Figures and Pleomorphism (as object detection) —child layers.

**Figure 3 diagnostics-13-02326-f003:**
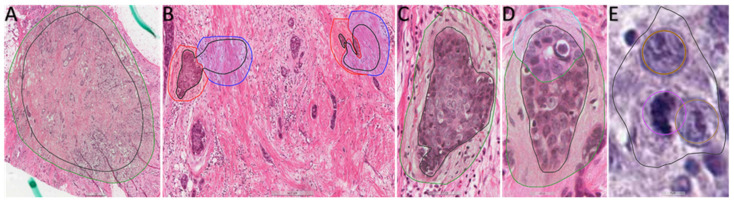
Extracts from WSI of the 10 ambassador slides uploaded in the Aiforia cloud showing examples of training regions (TR) and training annotations (TA). (**A**) TR and TA for Tissue detection, 4× magnification. (**B**) TR and TA for Normal Tissue (blue training annotations) and Tumor Tissue (red training annotations), 20× magnification. (**C**) TR and TA for Solid Component (green training annotations), 40× magnification. (**D**) TR and TA for Solid Component (green training annotation) and Tubule Formation (light blue training annotations), 40× magnification. (**E**) TR and object detection (brown for pleomorphism and purple for mitosis), 40× magnification.

**Figure 4 diagnostics-13-02326-f004:**
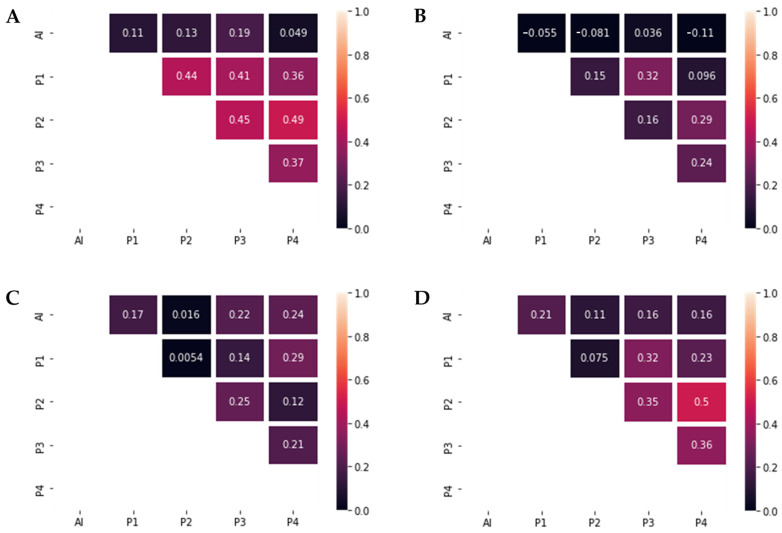
Graphic representations of Cohen’s κ. (**A**) Tubule grading agreement. Interobserver agreement between the pathologists was moderate (Cohen’s κ = 0.41–0.49), while the agreement with the DL model was slight (Cohen’s κ = 0.049–0.19). (**B**) Pleomorphism grading agreement. Interobserver agreement between the pathologists ranged from slight to fair (Cohen’s κ = 0.096–0.32). No significant agreement was present between the pathologists and the DLM (Cohen’s κ = −0.11–0.036). (**C**) Mitotic count grading agreement. Interobserver agreement between the pathologists ranged from slight to fair (Cohen’s κ = 0.0054–0.29), as well as the agreement between the pathologists and the DLM (Cohen’s κ = 0.016–0.24). (**D**) Nottingham Grade agreement. Interobserver agreement between the pathologists ranged from slight to moderate (Cohen’s κ = 0.075–0.5), while the agreement between the pathologists and the DLM ranged from slight to fair (Cohen’s κ = 0.11–0.21).

**Figure 5 diagnostics-13-02326-f005:**
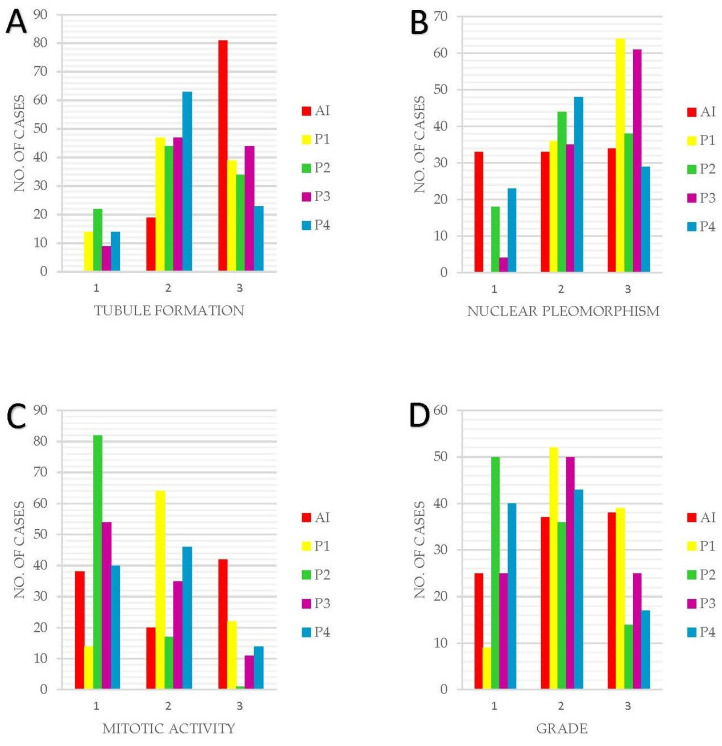
Graphic representation of the NGS component scores and final grading between AI and HA. (**A**) Tubule formation. (**B**) Nuclear Pleomorphism. (**C**) Mitotic Activity. (**D**) Grading. Overall, in this figure, it can be observed that the AI model tends to assign a higher grade to cases more frequently than a lower grade. This behavior is not a result of the model’s caution, but rather a reflection of its training, which emphasizes the identification of potentially severe cases due to the higher risk associated with underestimating the severity. Furthermore, for lower grade detections, the AI model’s grading aligns with the mean or average value of pathologist findings, indicating a balanced approach in these instances.

**Figure 6 diagnostics-13-02326-f006:**
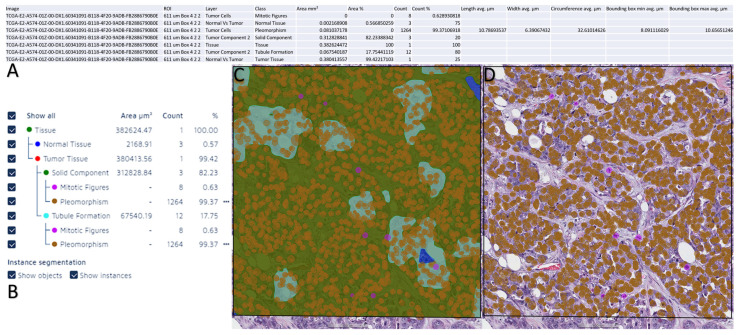
Data extraction from Aiforia Create and data visualization on Aiforia Create. (**A**) Summary of a WSI analyzed by the AI model. (**B**) Summary of the AI model detection on the layer model used. (**C**) Heat map of the AI model detection for all layers. (**D**) Heat map of the AI model detection for pleomorphism and mitosis.

**Table 1 diagnostics-13-02326-t001:** The number of cases with each tumor stage for the respective NGS. The percentage of the total number of cases for each NGS is shown in parentheses. The majority of tumors were stage IIA. All stage IIIB and IIIC tumors had an NGS of 3.

Grade	1 (N = 10)	2 (N = 25)	3 (N = 47)
Tumor Stage			
I	1 (10%)	4 (17%)	4 (8.9%)
IA	1 (10%)	1 (4.2%)	3 (6.7%)
IIA	0 (0%)	0 (0%)	2 (4.4%)
IIB	2 (20%)	8 (33%)	8 (18%)
IIIA	1 (10%)	3 (12%)	4 (8.9%)
IIIB	0 (0%)	0 (0%)	2 (4.4%)
IIIC	0 (0%)	0 (0%)	5 (11%)

**Table 2 diagnostics-13-02326-t002:** The number of cases with a particular score for each component. The percentage of total cases with each NGS is shown in parentheses. For grade 1 and grade 2 tumors, a tubule formation, nuclear pleomorphism score of 2, and a mitotic score of 1 were the most frequent. In grade 3 tumors, a score of 3 was the most frequent of all components.

Grade	1 (N = 10)	2 (N = 25)	3 (N = 47)
**Tubule formation**			
1	2 (20%)	0 (0%)	0 (0%)
2	8 (80%)	15 (65%)	1 (2.3%)
3	0 (0%)	8 (35%)	42 (98%)
**Nuclear pleomorphism**			
1	0 (0%)	1 (4.3%)	0 (0%)
2	10 (100%)	13 (57%)	7 (16%)
3	0 (0%)	9 (39%)	36 (84%)
**Mitotic activity**			
1	9 (90%)	12 (52%)	0 (0%)
2	1 (10%)	7 (30%)	10 (23%)
3	0 (0%)	4 (17%)	33 (77%)

**Table 3 diagnostics-13-02326-t003:** Tumor stage, NGS component scores, and NGS for each (were information was found) of the WSIs used in training the deep learning model (DLM). All ID information about the 10 training WSI used can be found in Appendix A.

Case ID	Tumor Stage	Tubule Formation	Nuclear Pleomorphism	Mitotic Activity	Nottingham Grade
TCGA-D8-A1JU	IA	2	2	1	1
TCGA-D8-A1XW	IIA	3	3	2	3
TCGA-BH-A0B7	IIB	2	3	3	3
TCGA-D8-A27L	IIIA	2	2	1	1
TCGA-D8-A1X9	IIB	2	2	3	2
TCGA-BH-A0H3	I	1	2	1	1
TCGA-EW-A2FS	IIB	3	3	3	3

**Table 4 diagnostics-13-02326-t004:** Performance of the DLM in segmenting WSI tissue elements. Precision, sensitivity, and F1 score were calculated as measures of performance. With regards to NGS components, tubule segmentation was the least accurate (precision = 94.95, sensitivity = 94.49, F1 score = 94.73).

All Values in Percentages	Total Area Error	False-Positive	False-Negative	Precision	Sensitivity	F1 Score	Training Loss Value
Tissue	0.46	0.13	0.33	99.85	99.63	99.74	0.0085
Normal Tissue	2.02	0.47	2.40	99.53	97.50	98.55	
Tumor Tissue	0.54	0.97	1.68	99.02	98.32	98.67	
Solid Component	5.06	0.97	6.73	98.97	93.27	96.04	
Tubule Formation	2.37	5.03	5.51	94.95	94.49	94.73	
Mitotic Figures	4.07	2.77	1.29	97.27	98.71	97.98	
Pleomorphism	3.51	1.50	2.00	98.49	98.00	98.24	

## Data Availability

All data is available from the corresponding author upon request. Data regarding the AI model is proprietary and belongs to Aiforia Technologies PLC.

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
