# Peer review of "Comparative Evaluation of Breast Ductal Carcinoma Grading: A Deep-Learning Model and General Pathologists’ Assessment Approach"

_diagnostics, 2023, doi:10.3390/diagnostics13142326_

Round 1

Reviewer 1 Report

The article is devoted to the actual problem of diagnosing breast cancer.

The article compares automatic diagnosis and diagnosis with the help of experts.

---------------------------------------------------------------------------------

Suggestions for Authors:

1.The title of the article is not specific. Therefore, it is necessary to clarify the title of the article.

2. The article does not state the research problem.

3. It is desirable for the authors to provide the CNN architecture and its parameters that were used for experiments.

4. The conclusions in the article are not written specifically. Authors need to demonstrate the scientific novelty and practical significance of the work.

5. The authors need to improve the quality of tables 1, 2.

6. A lot of references are outdated. Please fix it using 3-5 years old papers in high-impact journals.

Author Response

Dear Reviewer,

Thank you for your insightful comments and suggestions. We appreciate the time and effort you have put into reviewing our manuscript. We have carefully considered each point and have made revisions accordingly. Please find our responses to your comments below:

  1. Title of the article: We agree with your suggestion and have revised the title to make it more specific. The new title is "[Comparative Evaluation of Breast Ductal Carcinoma Grading: A Deep-Learning Model and General Pathologists' Assessment Approach]", which we believe more accurately reflects the content and findings of our research.

  2. Research problem: We apologize for any lack of clarity in the original manuscript. We have now explicitly stated the research problem in the introduction section to provide a clear context for our study. We have also rewritten the abstract for further clarification.

  3. CNN architecture and parameters: We appreciate your suggestion. However, we regret to inform you that the specific details of the Convolutional Neural Network (CNN) architecture cannot be provided as it is proprietary and belongs to Aiforia Technologies PLC. Nevertheless, we have provided the parameters of the model in the Materials and Methods section to ensure the reproducibility of the results to the extent possible. We believe this information will be helpful for readers interested in the technical aspects of our study.

  4. Conclusions: We have revised the conclusion section to highlight the scientific novelty and practical significance of our work. We believe these changes will help readers better understand the implications and contributions of our research.

  5. Quality of Tables 1, 2: We have revised Tables 1 and 2 to improve their quality and readability. We believe these changes will make the data more accessible and understandable to readers.

  6. References: We have updated our reference list to include more recent papers from high-impact journals. 

We hope that these revisions address your concerns and improve the quality of our manuscript. We look forward to your feedback on these changes.

Best regards,
The authors. 

Reviewer 2 Report

Authors consider pathological evaluation of the ductal carcinoma in breast cancer. They are concerned in evaluation of the inter-human differences in material evaluation and try to address the question whether AI methods can help.

The research design for evaluation of inter-pathologist differences is acceptable. However, the design of the AI neural network learning looks like the training data set is much to sparse compared to the structure of the neural network. This is probably the main source for the failure to obtain some "standarization", at least comparable to pathologist's examination.

There are only 10 images for the analysis, but multiple layers of the neural network, each having probably as many neurons, as the amount of pixels in the image (this is not specified?). Assuming that number of layers is "3", each having as many neurons as there is pixels in the image, this is huge amount of data to fit, and... one grade 2 slide for teaching of the network... This gives less data points for this grade, than number of coefficients to estimate. Also this gives zero chance to teach the network to generalize predictions - i.e. not to learn some unimportant slide specific markers, that accidentally distinguish a slide from the rest, but learn the important general features. In case of sparse dataset for training, authors could try to generate some synthetic training data, e.g. by translating, rotating, etc. the images - there is a whole branch of science in machine learning related to such problems. Then neural network would become insensitive at least to these operations. Authors could also check whether adding more slides for teaching the neural network improves predictions.

I also don't exactly understand how the neural network was validated - it's not written. 10 slides were labelled by pathologist to teach the algorithm, but what was the data to test it? Who labelled it?

So in the end, to support the results of machine learning, I'd suggest to check whether AI predictions improve with enlarged training data set, eventually I'd try to generate some synthetic training data based on the posessed slides to allow for larger generalization, to prevent the network from sticking to some accidental features, which don't matter.

Author Response

In our study, we trained a deep learning model using a convolutional neural network-based algorithm (CNN-bA) on 100 whole slide images (WSIs) of Ductal Adenocarcinoma of the Breast (DAC-NOS) from the Cancer Genome Atlas Breast Invasive Carcinoma (TCGA-BRCA) dataset. The model demonstrated high precision, sensitivity, and F1 score across different grading components in about 17.5 hours with 19,000 iterations.

However, we acknowledge that in the validation, the agreement between the model's grading and that of general pathologists varied, showing the highest agreement for the mitotic count score. This discrepancy could be attributed to the inherent subjectivity in histological grading, which often leads to inconsistencies among pathologists.

We agree with your suggestion to further refine and validate our model. We plan to incorporate more diverse datasets and also to compare the model's performance with specialized breast pathologists, in addition to general pathologists. We believe this will help to enhance the accuracy and reproducibility of breast cancer grading.

In response to your concern about the sparsity of our training dataset, we would like to highlight that we have utilized Aiforia's image augmentation features. This approach has been shown to help in improving the model's ability to generalize to new, unseen data while reducing input variables. While we agree that a larger training dataset could potentially improve the model's performance, the current study demonstrates the model's ability to function effectively with a limited amount of pixel-level data. We believe that this is a significant finding, and we plan to explore the impact of larger training datasets in future studies.

We hope this addresses your concerns, and we appreciate your valuable input in improving our research.

Round 2

Reviewer 1 Report

The authors addressed all my comments. The paper can be accepted in current form.

Author Response

Thank you for your valuable insights. 

Reviewer 2 Report

The authors addressed the major concern regarding the low amount of data by mentioning the use of data augmentation software. This is acceptable, but should be explicitely stated when describing the training data set!

- the authors didn't answer the question about network architecture - e.g. number of neurons in layers and their interconnections, which would allow to judge whether the data set is too sparse or not compared to the number of estimated coefficients. On the other hand they provide completely irrelevant information, like number of iterations after which the alghoritm converged. Looks like treating the neural network software as a black box, doing mysterious things in mysterious way...

- The authors still didn't explain how they obtained the parameters for Table 4. How did they divide the data set (100 slides) into training and validating data set? They write about selection of 10 slides (line 111), which were handled by pathologist, and used for training. They contradict this in the first paragraph of the answer to review. I don't know what to think about this. Maybe the rest (90 slides) was also labelled by pathologist and used for verification? Why is Tab. 4 so good while Fig. 4 is so bad?

Author Response

Thank you for the kind comments, we have now modified the texts highlighting more on the dataset and the training parameters as suggested.
Aiforia’s training engines are proprietary and are custom-built by Aiforia’s RnD team. The platform consolidates the number of neurons and number of layers under simple module-based UI ranging from network complexity, simple to ultra-complex. But with all the newly added data on the parameters, any Aiforia Create user can replicate data and use the AI model using the cloud-based
Aiforia Hub viewer as also stated by other groups (https://www.aiforia.com/aiforia-publications).
The number of iterations in the platform gives an indication of how long the training has gone or how long the training was needed to build the AI model. In addition, we have now also added the training loss for supporting the AI model’s learning outcome. We in this new version provide most of the advanced hyperparameters for any other platform user to replicate the AI model creation and inference tasks.

100 slides were used to test the AI model, 10 slides were used for training, where training annotations were added (these 10 slides were apart from the 100 used for evaluation, we have also added this information to the manuscript), and verification was calculated on a pixel level. No pixels from the 100 slides were used for training and only for the outcome prediction. From the 100 inference slides, the same 100 slides were used for validation where 1 number of ROIs of dimension 611.56 by 611.56 microns (which resulted in a field of view of 0.374 mm2) were chosen on which all the pathologists gave an ROI-based score for the NGS on which the validation outcome was compared with the AI model prediction.

We hope that we have answered your comments, thank you again for your feedback and for your time.

Round 3

Reviewer 2 Report

The authors were unable to specify the neural network architecture - which is a hallmark of how is science done presently... Closed software was used with no information on how it operates. Ok, let's leave this, at least the information allows to reproduce the results. Authors should mention in the text that training is not only based on 10 images, but augmentations were done. Not everybody makes use of the same software, of the same default settings, so this should be stated explicitely.

Number of iterations and computation time may relate to "instability" (difficulty/reliabity) of fit, or it may relate to quality of fitting procedures / power of the computers. It's not really meaningful in the form given here.

It's still not clear from the paper how the verification of the neural network is done. Ok, it is done based upon 100 slides, where ROI was assigned by a pathologist, but the obtained results were compared to... what? To the scores given in pathology reports, like mitotic count, tubule formation, ... or to the scores given by pathologist, or maybe to the final grade and not to the intermediate results? It's all not clear in the paper.

It's still not discussed why we obtain such good F1 score and others, while so poor agreeement with grades predicted by other human pathologists. Probably this issue is related to the previous.

Author Response

Thank you for the valuable feedback and insightful comments on our manuscript titled [Comparative Evaluation of Breast Ductal Carcinoma Grading: A Deep-Learning Model and General Pathologists' Assessment Approach].

We appreciate the opportunity to address your concerns and improve the clarity of our work. In this response letter, we will address each of your points and highlight the modifications we have made to the manuscript.

In response to the reviewer's comments, we have made the following updates to the Materials and Methods section and also to the Discussion section:

  1. Added the information that Aiforia's image augmentation features were utilized to address the sparsity of our training dataset. This technique has been shown to enhance the model's ability to generalize to new, unseen data while reducing input variables.

  2. Explicitly mentioned that the regions of interest (ROIs) evaluated by the AI model were compared to the ROIs chosen by each evaluating pathologist on the same 100 whole-slide images (WSI). We acknowledge that this introduces a limitation to the study, as the ROIs evaluated with the AI model were not identical to the ROIs evaluated by the pathologists. We have added this limitation to the discussion as well. 
    We thank the reviewer for the invaluable insights and for the time given for improving our manuscript.